# ‘Stolen Time’—Delivering Nursing at the Bottom of a Hierarchy: An Ethnographic Study of Barriers and Facilitators for Evidence-Based Nursing for Patients with Community-Acquired Pneumonia

**DOI:** 10.3390/healthcare9111524

**Published:** 2021-11-09

**Authors:** Signe Eekholm, Karin Samuelson, Gerd Ahlström, Tove Lindhardt

**Affiliations:** 1Department of Health Sciences, Faculty of Medicine, Lund University, P.O. Box 157, SE-221 00 Lund, Sweden; karin.samuelson@med.lu.se (K.S.); gerd.ahlstrom@med.lu.se (G.A.); tove.lindhardt.damsgaard@regionh.dk (T.L.); 2Department of Internal Medicine, Copenhagen University Hospital, DK-2900 Hellerup, Denmark

**Keywords:** evidence-based practice, fundamental care, nurse manager, nursing care, nursing practice, organisational behaviour, patient safety, work organisation

## Abstract

The research has reported a high prevalence of low-quality and missed care for patients with community-acquired pneumonia (CAP). Optimised nursing treatment and care will benefit CAP patients. The aim of this study was to describe the barriers and facilitators influencing registered nurses’ (RNs’) adherence to evidence-based guideline (EBG) recommendations for nursing care (NC) for older patients admitted with CAP. Semi-structured focus group interviews (*n* = 2), field observations (*n* = 14), and individual follow-up interviews (*n* = 10) were conducted in three medical units and analysed by a qualitative content analysis. We found a main theme: ‘‘stolen time’—delivering nursing at the bottom of a hierarchy’, and three themes: (1) ‘under the dominance of stronger paradigms’, (2) ‘the loss of professional identity’, and (3) ‘the power of leadership’. These themes, each comprising two to three subthemes, illustrated that RNs’ adherence to EBG recommendations was strongly influenced by the individual RN’s professionalism and professional identity; contextual barriers, including the interdisciplinary team, organisational structure, culture, and evaluation of the NC; and the nurse manager’s leadership skills. This study identified central factors that may help RNs to understand the underlying dynamics in a healthcare setting hindering and facilitating the performance of NC and make them better equipped for changing practices.

## 1. Introduction

Effort has been made worldwide to develop evidence-based guidelines (EBGs). EBGs are important aids in translating scientific evidence into daily practice [1]. They have been developed to support healthcare professionals (HPs) in decision-making regarding appropriate and effective treatment and care [1]. However, previous studies have indicated that there is a variety in practice management, and the performance of treatment and care is often inconsistent with EBGs [2,3]. Particularly, fundamental nursing care (NC) interventions are performed haphazardly, unsystematically, and, in the worst cases, are missing [2,3,4,5]. Therefore, there is a need to explore what hinders RNs in performing optimal and safe treatment and care and what will support them in implementing EBG in daily clinical practice.

A group of patients who appear to be deprived of NC interventions according to EBG are older patients with community-acquired pneumonia (CAP) [2,3,6], leading to fatal patient outcomes. CAP is an acute inflammatory lung disease and a significant cause of morbidity and mortality among older persons (>65 years) [7], thus representing a major cost and capacity challenge for hospitals and society [8,9].

Studies have shown the unsuccessful transfer of EBGs into routine clinical practice for patients with CAP regarding diagnostic procedures, medical treatment, and NC, of which NC appears to be the most neglected [6]. According to EBGs [10,11], NC for patients with CAP consists of interventions such as respiratory therapy, sputum mobilisation, oral care, mobilisation, fluids, and nutrition therapy. Although not complex or highly technical interventions, they address fundamental needs and are essential NC for older patients with CAP. The crucial role of systematic, evidence-based NC delivered by registered nurses (RNs) for patient safety and economy has been established through extensive research over decades. Studies have shown positive effects on hardcore patient outcomes such as morbidity, mortality, length of stay, and readmission rates [12,13,14,15,16]. However, researchers consistently report a high prevalence of missed or undone NC in clinical practice [4,17].

The unsuccessful transfer of evidence into practice presented in previous studies [4,6,18] raises the question of what hinders RNs in adhering to EBGs, particularly in light of the fatal consequences for patients and the economic and capacity burdens for healthcare settings. According to implementation researchers [19,20], context-related barriers inhibiting the implementation and facilitators for the implementation of EBGs in clinical practice must be explored [21]. Systematically exploring the barriers and facilitators and their rationale at multiple levels (individual, group or team, organisation, and social contexts of care provision) and using the knowledge to develop implementation strategies and interventions are needed to increase the evidence uptake in clinical practice [19]. Therefore, this study aimed to describe work-based barriers and facilitators at the individual, team, and organisational levels influencing RNs’ adherence to EBG recommendations for NC for older CAP patients.

## 2. Materials and Methods

### 2.1. Design

An ethnographic design was used. The ethnographic approach is appropriate for groups of persons in their natural setting with commonalities [22,23,24]. This approach is of value to researchers exploring common behaviours, experiences, shared features, or patterns of individuals in a bounded group and distinct situations or issues within a specific context [22,23,24]. In our study, this applies to the participants in their everyday work as RNs in a hospital setting treating and caring for patients with CAP. This approach is characterised by the researcher’s role as an instrument and allows the researcher to collect data using multiple methods. A central belief in ethnography is that people’s behaviour can only be understood in its context, and the elements of human behaviour cannot be separated from their context within its purpose and meaning. Thus, context is essential for understanding the phenomenon of interest. Therefore, data were collected by field observations and individual follow-up interviews, where the first author (SE) became part of the specific context within which the RNs operated in order to achieve a deeper understanding of the studied phenomenon.

The researcher’s preunderstanding is an important precondition for discovering different aspects of the phenomenon, although it may also cause limits. Hence, it is important for the researcher to expand their preunderstanding while staying curious, open-minded, and sensitive to the experiences and phenomena that contradict their expectations and prejudices [25]. The researcher’s preunderstanding in this study was constructed by many years of clinical experience as a clinical nurse specialist and her expert knowledge of the treatment and care of patients with CAP. To enhance the awareness of the phenomenon under study, she expanded her preunderstanding by studying the literature about EBG recommendations for CAP [10,11] and the barriers to and facilitators for HPs’ adherence to EBGs [3,6]. Furthermore, the first author’s work was critically supervised by the three other authors, each with many years of clinical experience and expert knowledge of performing ethnographic research. To enhance the interrater reliability of the findings, all the authors were involved in continuous critical reflections on shared meanings, thus expanding the first author’s preunderstanding of the factors influencing HPs’ behaviour in clinical practice.

The Theoretical Domain Framework (TDF) [26] (14 domains selected from 33 behaviour change theories and 128 constructs) was central in this process, as it captures barriers and facilitators for HPs’ behaviour at the individual, team, and organizational levels. The 14 TDF domains consist of: ‘Knowledge’, ‘Skills’, ‘Social/professional role and identity’, ‘Beliefs about capabilities’, ‘Memory, attention and decision processes’, ‘Beliefs about consequences’, ‘Environmental context and resources’, ‘Social influences’, ‘Intentions’, ‘Optimism’, ‘Goals’, ‘Behavioural regulation’, and ‘Reinforcement’ (Table 1). These domains constitute an extensive framework for potential factors that might influence clinical behaviour. The TDF was initially developed to understand HPs’ behaviours related to the implementation of EB recommendations. It has been widely used to assess barriers to and facilitators for implementation problems and provides a useful basis for assessing multiple perspectives (e.g., RNs, managers, and interdisciplinary teams of HPs) by using multiple sources of data, thus increasing the validity of the findings. The knowledge of these findings can be used to design tailored theory-informed implementation strategies that will support changes in a clinical practice.

### 2.2. Setting

This study was conducted in a department of internal medicine in a Danish university hospital. In this department, approximately 10% of acutely admitted patients (≥65 years) are diagnosed with CAP in the emergency department and, when needed, admitted for further treatment and care at three medical units (58 beds): infectious diseases (21), respiratory diseases (22), and a short-term unit (15). These three medical units participated in this study.

The nursing staff comprised 70 employees (17 licensed practical nurses (LPNs) and 53 RNs). The management consisted of one head nurse, two nurse managers, and three assistant nurse managers. In one unit, the nurse manager position was vacant. Two medical units had a clinical nurse specialist and two physiotherapists. Three to four physicians consulted each unit daily. A secretary was employed to help the staff with administrative tasks. A number of cleaning and kitchen staff assisted the units every day.

All three units were similar in structure and organisation. Each unit employed both LPNs and RNs, organised in teams consisting of one LPN and one RN or only RNs. Each team conducted direct NC for approximately 6–8 patients where one RN was in charge of interdisciplinary cooperation and coordination and, as such, functioned as a kind of ‘group leader’. However, this structure could differ from day to day, depending on sick leave. On days with sick leave, two RNs could be responsible for 12–14 patients each, and four LPNs took care of 22–26 patients. Patient plans were made in cooperation between physicians, RNs, LPNs, a nurse manager, an assistant nurse manager, and physiotherapists at daily interdisciplinary meetings and during patient rounds in cooperation with the patient.

### 2.3. Sample

The participating RNs from the three units were selected by purposeful sampling. Eligibility included RNs taking care of at least one patient diagnosed with CAP (≥65 years) at the beginning of the observation. The researcher presented herself to all the patients who were part of the observations and informed them about the study. All patients provided consent to participate. Once the patient was identified, the first author approached the RNs and asked for their consent to participate in the study. As the RNs worked either alone or in close teams with their RN colleagues and LPNs, the LPNs were asked for consent, and the whole team taking care of at least one patient with CAP was observed, though with a main focus on the RNs. Fourteen RNs were included in the study. The RNs and LPNs had the right to refuse participation, but none did. However, four of the 14 RNs could not participate in the individual follow-up interviews because of a busy work situation. During the observation, the RNs interacted with 88 interdisciplinary HPs (LPNs; physicians; physiotherapists; nurse managers; assistant nurse managers; clinical nurse specialists; and municipality, kitchen, and cleaning staff).

### 2.4. Data Collection

The data were collected from November 2017 to March 2018, starting with two semi-structured focus group interviews (*n* = 6 RNs in each group), followed by field observations (*n* = 49 observation hours) and individual follow-up interviews with RNs (*n* = 10) conducted immediately after the observations. Data collection was guided by (1) EBGs’ criteria for nursing interventions related to CAP [10,11,27,28,29,30,31], (2) the TDF [26], (3) data from a previous descriptive study [6] disclosing gaps between evidence-based recommendations for NC and the current clinical practice in the same units, and (4) the researcher’s preunderstanding based on expert knowledge from many years of experience as a clinical nurse specialist in the department of internal medicine. The overview of the data and data collection is presented in Table 2.

#### 2.4.1. Focus Group Interviews

Two focus group interviews (six RNs in each group) were conducted in the units in an undisturbed office by two of the authors: the moderator (first author, SE) and the observer (last author, TL). The moderator briefed them about the aim of the interview and led the interview by following a semi-structured interview guide. The interview guide (Appendix A) was developed based on the TDF framework of 14 domains [26] to describe the barriers and facilitators influencing the RNs’ adherence to the EBG recommendations for NC. The interview contained questions such as; Do you know, or have you read the content of EBG for patients with CAP? Do you know NC interventions for patients with CAP and how to perform them? To what degree do your colleagues, context, or resources influence your performance of EBGs recommendations for NC for patients with CAP? The questions were planned but were flexible, providing the researcher an opportunity to change the sequence of questions and probe for more information. Further, the participants were encouraged to discuss the questions among themselves. The interview sessions lasted approximately 1.5 h, were digitally recorded, and were transcribed verbatim.

#### 2.4.2. Field Observations

The focus group participants indicated time pressure as an explanation for missing care. However, they were unable to account for how they used the available time, entailing the need for a further exploration of RNs’ prioritisation of time and tasks by conducting field observations. In-line with the ethnographic approach, the RNs were observed during their full day shifts. Observations were carried out by placing the first author (SE) on the side-lines of the RN’s activities related to the treatment and care of patients with CAP. Field notes were taken continuously during the observations to provide a detailed description of the observed situations. The field notes included information on the date, place, time of the observations, environment, the participants, verbatim verbal exchanges, and personal reflections in chronological order of what happened during the observations. Data collection was completed when it was deemed that a comprehensive picture of the influencing barriers and facilitators for adherence to EBGs was attained. In total, the observations lasted 49 h.

#### 2.4.3. Individual Follow-Up Interviews

Individual follow-up interviews were performed with the RN immediately after observations at the end of the shift and took place in an office in the hospital unit. The interviews were conducted to deepen the understanding of the barriers and facilitators influencing RNs’ adherence to EBGs for NC, guided by the issues or reflections that arose during the observations. After each interview, the RNs were asked not to reveal the study aim or issues to their colleagues to prevent future participants from modifying their normal patterns of clinical practice during the observations. The interviews were audiotaped and transcribed verbatim. Interviews ranged from 30 to 45 min.

### 2.5. Analysis

The transcribed field notes and interview texts were analysed using a qualitative manifest and a latent content analysis [32]. The manifest content is the descriptive part of the analysis in which the surface structure of the text is revealed, whereas the latent analysis involves an interpretation of the underlying meaning of the text [32]. Initially, all authors read the transcription of the first two observations and interviews to gain an understanding of the data. The first and last authors read all the transcribed texts several times to obtain a deeper overall understanding. This was a continuous iterative process going back and forth between the field notes and interview data as new understandings emerged. The transcribed text was subsequently analysed using open coding. Both authors wrote memos, reflections, and interpretative attempts for the first two transcribed texts with empirical knowledge of the TDF domains, the EBG criteria for nursing interventions, and their preunderstanding of the context and clinical experience in mind. The two authors discussed and compared the analysis until a common understanding of the texts and the analysis process was reached. The first author analysed all the transcribed texts individually and divided the text into meaning units related to the study aim. The meaning units were labelled with a code, sorted into subcategories, and categorised according to the manifest content. The last author critically reviewed the meaning units, codes, subcategories, and categories and discussed them in depth with the first author at several meetings to adjust the system of subcategories and categories to establish a hierarchy of the manifest content. Subsequently, the latent content analysis was performed by the first and the last author and involved searching for the underlying meaning on an interpretative level across the categories and subcategories, as well as in the meaning units and codes and the above-mentioned memos and notes. A hierarchy of the subthemes, themes, and main themes emerged from this process, expressing the latent content of the text.

All four authors critically questioned and discussed the findings until a consensus was reached and small adjustments were made. Furthermore, the included participants were presented the findings to increase their trustworthiness. They recognised the content and the thematic patterns and approved the findings.

To ensure transparency, the reporting of the manuscript followed the Standards for Reporting Qualitative Research (SRQR) checklist (Appendix A).

## 3. Results

One main theme emerged from the content: ‘‘stolen time’—delivering nursing at the bottom of a hierarchy’, capturing three interrelated themes: ‘under the dominance of stronger paradigms’, ‘the loss of professional identity’, and ‘the power of leadership’. The themes consisted of two to three subthemes (Table 3).

All the themes and subthemes presented barriers and facilitators influencing RNs’ adherence to EBGs recommendations for NC at the individual, team, and organisational levels. These barriers, facilitators, and their interrelatedness are presented in Figure 1.

### 3.1. ‘Stolen Time’—Delivering Nursing at the Bottom of a Hierarchy

The content of the main theme ‘‘stolen time’—delivering nursing at the bottom of a hierarchy’ expressed that RNs lacked time to perform evidence-based NC. They did indeed lack time; however, this was not because of the workload. The observations illuminated that time for NC was lacking because it was stolen. RNs’ time was stolen by other professions, management, colleagues, their organisation of work, and even by the RNs themselves, for example, when they lacked the competence to work systematically with a focus on the patients’ fundamental needs. Furthermore, the RNs lacked professional identity and terminology, as well as power and leadership, to prevent other professions from stealing their time. Therefore, the RNs helplessly ended up at the bottom of the hierarchy. Consequently, NC was carried out unplanned and unsystematically in the time gaps between the demands from other professions.

#### 3.1.1. Under the Dominance of Stronger Paradigms

This theme illuminates that the context, a working culture organised around the biomedical model and cooperation with other professions, strongly influenced the RNs’ focus, their use of time, and prioritisation of NC, placing NC interventions lowest in the hierarchy of tasks in these units with the acceptance of RNs and nurse managers.

##### Detained by the Medical Model

The observations revealed that the RNs’ working processes, focus, and planning of their tasks were strongly influenced by the context in which the medical model ruled with its biomedical focus. NC was invisible in interdisciplinary meetings and communication, where the attention was always on the other professions’ tasks and goals. For example, RNs attended compulsory interdisciplinary meetings with a high relevance for physicians, mainly focusing on planning patient flow, medical aspects, or diagnostic procedures. At those meetings, every patient’s treatment plan was discussed and revised because of the patient’s health and recovery status. The RNs prepared themselves daily for those meetings by extracting biomedical information from the patient records and served physicians and other professionals with information about drug therapy, blood exams, and patient flow instead of focusing on NC:
*The RN presents the report focusing on whether samples have been taken or not, what the blood samples showed, and what medication patients get. There is only a focus on medical problems.**(Observation 6, line 115–119)*

These meetings directed the RNs’ focus and planning of their day shift program towards the medical, administrative, and physical and biomedical aspects of care rather than NC and the patient’s fundamental needs. This appeared to be one of the greatest barriers to performing NC in a systematic, person-centred (PCC), and knowledge-based way.

Moreover, the RNs expressed that carrying out tasks in connection to the patient flow was extremely important, as nonadherence to physicians’ plans for the patient flow caused delayed discharges and increased expenses for the unit. Lack of time was considered the cause of any possible nonadherence. Nonadherence to physicians’ discharge plans upset and frustrated the RNs and physicians, and the RNs felt guilty if patients were not discharged according to the plan.


*The physician speaks very strictly, and there is no doubt that she is annoyed. You can see that, in her body language and hear it in her voice. In addition, the RN is snarled at and accused of not having contacted the municipality on behalf of one of the patients.*

*(Observation 6, line 104–114)*


Furthermore, observations revealed that RNs used their time to plan the patient rounds for the physicians but seldom attended them, as they used the time to catch up with other tasks. In general, RNs expressed that they had to concentrate on administrative and biomedical tasks and check on the physicians’ tasks, as they felt responsible for the patient’s condition and stability, and they were to blame if something was overseen or went wrong.

##### Time Thieves

There was consensus among the RNs that they lacked time and that this was due to a heavy workload. The disproportionate relationship between time and tasks was highlighted as the main barrier for delivering systematic and evidence-based NC. Indeed, the participating RNs were busy, and time seemed scarce; however, the observations revealed that time was there but was stolen by other professionals in the form of interruptions and unannounced visits. RNs were available 24 h a day, and they could be interrupted anytime and anywhere. Additionally, their lunchbreaks often lasted five to ten minutes.

Even the kitchen staff had influence over the RNs’ time. The delivery of meals from the hospital kitchen was organised according to the kitchen’s work schedules and had a very tight timeframe that allowed the RNs only half an hour to distribute the meals to their patients before the carriages were taken back to the central kitchen. This left little or no time for patient participation or person-centred care, let alone nutritional information, or to encourage the patient to eat. This was accepted as the norm and never questioned at any level in the team.

The lack of overall coordination of the many professional groups’ workflows and work procedures constituted a fundamental condition and a barrier for the delivery of nursing. Consequently, many tasks were organised at the same time. The above-mentioned work schedule of the hospital kitchen is one example, which provided four to six RNs half an hour to serve the meal, administer medicine, and mobilise 15–25 patients to the dining area, while being interrupted by colleagues, patients, nurse managers, or other professionals:
*While the RN is distributing the meal: The physician contacts her and asks if a blood test for blood culture has been taken. The RN begins straight away to investigate this. The RN sets out to find the physician to pass on the message. While the RN is back to serve the meal to her patients, the physician contacts her again and asks her to order new blood tests (a task that physicians normally would do).**(Observation 1, line 430; 440; 447–448)*

The hospital electronic patient record (EPR) was a time thief in itself, although this seemed to be a consequence of lacking competence at the user level. The RNs felt that they received insufficient training to use the EPR and, therefore, spent a large amount of time reading and extracting information and used the EPR differently. This resulted in insufficient knowledge bases for treatment and care plans, as well as duplicated working processes. Homemade pocket sheets were filled in by hand by each RN at the start of every shift, although the same information could be extracted from the EPR. RNs acknowledged that it was a duplicate work but had no other solutions for a more efficient working process.

#### 3.1.2. The Loss of Professional Identity

This theme illuminates that the context and other professionals influenced RNs’ focus and their use of time. In addition, the lack of professional identity and professionalism was a barrier at the individual level influencing RNs’ performance in NC.

##### Filling the Leftover Time Gaps

Working under time pressure and not being in charge of their own time left RNs to deliver NC in the time gaps that were left over from the demands of other professions. The time gaps were small amounts of time delineated by the prior and next tasks (coordinating or biomedical) or interruptions. When and where these time gaps would occur was uncertain; hence, RNs prioritised at the spur of the moment, and NC was delivered unsystematically and haphazardly. NC and time-consuming tasks were also under-prioritised or left undone, although RNs were painfully aware of not being able to perform optimal evidence-based NC and the consequences of missing care for patients. However, they seemed powerless and lacked solutions to this problem.


*I don’t have time for it all. I must learn that there isn’t time for me to do everything. I try to choose what I can (prioritize). The problem is that even though I haven’t eaten breakfast and stuff like that (she means, skipping her break times), there’s still not enough time. I want to do it all, but … what can I do? I can’t (she sighs resignedly).*

*(Interview 11, line 893–902)*


In most cases, tasks left undone were handed over to colleagues during shift changes, hoping that they had time. In several observations, these tasks were still incomplete the following day.

Furthermore, colleagues or other professionals requested NC interventions from the RNs, e.g., giving patients fluids, nutrition, or respiratory therapy. This indicated an unsystematic organisation of care, possibly due to a lack of professional identity or role clarification. In several cases, nurse managers or physicians asked RNs about the fundamental NC interventions during the interdisciplinary meetings or during the patient rounds; they did not have a clear answer. However, RNs were well-informed and well-prepared for questions regarding patient haemodynamics or medication plans. The confusion around role clarification was visible during the actions and communication between RNs and physicians. RNs could guide physicians about tasks that were specific to their profession (e.g., the ordination of medication, blood tests, or discharging patients in the IT system), and physicians guided or informed RNs on how to conduct fundamental NC tasks (mobilisation, seating, and respiratory treatment). In addition, LPNs asked RNs to conduct tasks that were specific to their profession while helping RNs administer medication.

##### Knowledge without Action

In general, RNs are knowledgeable about nursing interventions such as oral care, respiratory therapy, sputum mobilisation, nutrition, and fluid therapy and mobilisation, which are recommended in EBGs for patients with CAP. However, observations illuminated that RNs often did not apply their knowledge in clinical practice. For example, patients did not receive oral care based on several observations. In the interviews about missing care, several RNs admitted that they were aware of the importance of oral care, but they often forgot about oral care when organising their tasks, or they prioritised other tasks instead:


*I think we all underprioritize oral care. I think that’s stupid because … I know how important it is. But I almost never think about it. I just forget. Yes. I just forget it. (RN looks down at the floor and her hands settle into her lap. Her body language signals embarrassment).*

*(Interview 2, lines 717–744)*


RNs delegated most of the fundamental NC to LPNs, while RNs took care of patient rounds, administrative tasks, medicine administration, the admission or discharge of patients, and attended compulsory meetings. Only when there was time or when LPNs needed help did RNs take part in fundamental NC. RNs expected LPNs to have the relevant and necessary knowledge and competence to carry out NC and for them to ask for help if needed. However, the observations revealed that this was not always the case, and the phenomenon ‘action without knowledge’ appeared. In several cases, RNs and LPNs performed interventions without knowledge of the interventions’ rationale, the recommended intensity, frequency, or effect of the interventions. For example, one of the RNs attended an acutely ill patient in need of oxygen therapy. Although she acted immediately by providing her patient oxygen, the level of oxygen was not sufficient according to the EBGs’ recommendations.

In another example, one LPN motivated and guided the patient to use positive expiratory pressure (PEP) therapy for sputum mobilisation. Communication between the patient and LPN revealed that the LPN was not aware of the correct technique or the intensity of the PEP usage. The observation and interview revealed that the LPN did not receive supervision by the RN and had no patient care plan to guide them in providing correct PEP therapy, consequently putting the patient at risk of treatment failure.

When delegating fundamental NC interventions to LPNs, RNs also passed on their responsibility. RNs expressed that they seldom had an overview of the care their patients did or did not receive.


*SE: What about nursing care? Do you know if it has been done? RN: I had a partner (she means LPN) … but I have not had an overview … I had my concentration mostly on tasks from the patient-round. Contact with the municipality and tasks like that.*

*(Interview 8, lines 505–509)*


The observations revealed that no supervision of the LPNs took place, and little feedback was delivered from the LPNs to the RNs regarding fundamental care. They sometimes communicated in passing, when meeting in the hallway, about how far they were with their tasks and arranged to help each other during available time gaps.

##### The Impact of a Professional Terminology

The RNs’ documentation and their communication with colleagues and other professionals revealed a lack of professional and concise terminology. This constituted a potential hazard to patient safety, as they had difficulty catching the attention of team members from other professions when arguing for the patient’s case. In several cases during the observations, this led to a delay in acute treatment, thus depriving the patient of timely intervention. The teamwork and the working climate were influenced by this, and sometimes, the RNs felt rejected and treated disrespectfully:
*RN: It’s been difficult. Today, it has been difficult. Ehhh … I think … ehhh, … that the doctors seem a little dismissive. In their communication. It is hard … to get in touch with them (tears in her eyes). SE: You got rejected? RN: (laughs a little) … you must have some really strong arguments and that’s probably where we’re missing … that it is probably there I … that it will slip if you do not have your arguments in place. Then you may not quite get your message through.**(Interview 1, lines 593–614)*

Further, RNs expressed that the lack of professional terminology resulted in seldom reading notes written by RNs when extracting information from patient records or preparing for interdisciplinary meetings. Several RNs found nursing notes irrelevant or lacking sufficient information about the patient’s care and treatment. Nursing notes even had a nickname: ‘cosy notes’. Consequently, nursing documentation was not used for planning and organising NC:
*SE: Do you read medical notes only? (I ask because I can see her skipping nursing notes). RN: No not only, but yes mostly … SE: Do you use patient care plans? RN: Not very much.**(Observation 1, lines 95–98; Interview 4, lines 431–441)*

Due to the lack of NC plans, RNs had to read physician notes to create an overview of the patient, patient status, and treatment plan. The prioritisation of physicians’ notes was also explained by the fact that they needed to be prepared for interdisciplinary meetings where they had to answer questions such as medical treatment, the patient’s medical status, and so forth. This seemingly blocked the view for fundamental NC, as the RNs’ focus turned to the physical and biomedical aspects of patient treatment. However, RNs who used professional terminology and who used evidence-based knowledge as arguments received the physicians’ attention and were treated respectfully.


*Interdisciplinary whiteboard meeting: The RN presents her patients by name, diagnosis, age, and the NC plan. She knows her patients and answers all questions from the physician and the nurse manager. She is professionally well-articulated. I (SE) get the feeling that she is respected and approached in a completely different way than her colleagues.*

*(Observation 1, line 236–247)*


#### 3.1.3. The Power of Leadership

This theme illuminates that nurse managers appear to have the power to mediate the working culture in the units and to support or eliminate the confining structures of the organisational hierarchy.

##### The Mediator of Culture and Hierarchy

In the units where the nurse manager was absent or focused on biomedical tasks rather than NC, nursing practice was task-oriented and focused on biomedical tasks. In these units, the hierarchy was most apparent, as in other professions’ schedules, organisation of the work and needs overruled RNs’ working process and obligation to deliver NC. Seemingly, NC was invisible and not integrated into the organisation of teamwork, as was also the case for the time needed to perform it. Moreover, none of the RNs or nurse managers openly delineated the boundaries for NC, thus claiming time and manpower for it.

Nurse managers can even contribute to the theft of time from NC. This was the case when the nurse managers had difficulty organising the RNs during a shift or were less resolute about their decisions. In those cases, RNs overruled the managers’ decisions of work organisation and spent time (in some cases, a whole shift) reorganising the plans laid out by the manager. In one observation, the RNs had difficulty in arriving at an agreement about how to organise the shifts and tasks amongst each other, and this resulted in a conflict. These situations could also occur if the nurse manager was absent during the morning shift change, where the decisions about work organisation were usually led by nurse managers. Consequently, RNs fell behind their working schedules and lost time on NC.


*While the RN from the night shift gives her report, the staff begins to discuss who has cared for which patients and which patients they should care for … The discussion continues, but no agreement was found … The RNs find their computer, but the discussion comes up again and one of the RNs notes the distribution on the whiteboard. There is no management present, her office is dark, and the door is closed.*

*(Observation 1, lines 28–30, 32–33, 37–39)*


##### Stealing Back Time

In contrast, one nurse manager had a consistent focus on NC and the power to put NC on the agenda in interdisciplinary cooperation, thus making nursing visible. She demanded RNs to be in charge and actively participated in the patients’ care planning process, with a focus on NC. She attended all the interdisciplinary meetings and continually demanded a status of the NC plans for every patient in her unit. In this way, she indicated the importance of NC and positioned it as an equal part of the overall patient treatment and care plan.


*The nurse manager evidently knows all the patients in the unit. Their status needs and their plan. She contributes with information when physicians ask questions the RNs cannot answer. At the same time, she asks RNs about the NC tasks that need to be done or tasks that have not been carried out. If RNs are in doubt, she makes suggestions on what to do or how to take action. She also asks the physicians about interventions such as nutrition, mobilisation, etc.*

*(Observation 5, lines 118–124)*


In addition, RNs who focused on biomedical tasks were asked to turn their focus back to NC. This nurse manager had a strong ally in the clinical nurse specialist, who was a skilled facilitator for improving NC by stimulating RNs to think and work based on evidence. The nurse manager and the clinical nurse specialist had daily meetings with RNs and LPNs where the evidence for NC interventions, relevant for the unit specialty, was presented and reflected upon. RNs participated actively in those meetings, either by presenting evidence or by discussing the implementation of evidence in their own clinical practice. Hence, the professional management of nursing in this unit had the power to steal back time for NC by mediating with RNs for their professional identity and possibly facilitating their ascent from the bottom of the hierarchy.

## 4. Discussion

The results revealed only a few facilitators, mostly at the organisational level, whereas barriers interrelated with each other were identified in all the themes. Furthermore, in multiple cases, we found exceptions where determinants could emerge both as a barrier and opposite, as facilitators (e.g., professional terminology, when lacking) emerged as a barrier, whereas the use of professional terminology emerged as a facilitator.

### 4.1. Individual Barriers and Facilitators

A central barrier at the individual level for RNs’ adherence to EBGs lacks professionalism. This appeared in the lack of professional terminology in verbal and written communication (e.g., the patient record). It also appeared in the absence of knowledge and skills and in an unclear role perception that influenced RNs’ ability to focus on, prioritise, plan, and perform fundamental and evidence-based NC. The concept of professionalism has been reported to reflect knowledge, intellectual and individual responsibility, autonomy, and collaboration that influence nurses’ practices, decision-making, knowledge sharing, and interprofessional collaborations [33,34]. The majority of the RNs in our study were knowledgeable about evidence-based NC interventions for patients with CAP and their underlying rationale, yet they had trouble applying this knowledge in practice and providing NC focus and attention. They were not able to use their knowledge due to contextual barriers, indicating a weak professional identity. This phenomenon was also reported by Voldbjerg et al. (2017) [35] in an ethnographic study of newly graduated RNs (*n* = 9), who unlearned their academic skills when entering clinical practice, in line with the findings of our study. The authors found that the context and working culture in a hospital setting appeared to have an impact on professional nursing practice.

Our results further indicated that a lack of professional identity and unclear role perception among RNs contributed to both the position of NC at the bottom of the organisational hierarchy and to the suboptimal quality in NC. In contrast, some RNs had a clear perception of their professional roles and boundaries, using evidence-based knowledge in their practice and communicating using professional terminology. This seemed to facilitate interdisciplinary collaboration, in that they were respected and caught the attention of their colleagues and the interdisciplinary team. Furthermore, the patients received optimal and timely treatment and care. Compared to our results, Bunkenborg et al. (2013), who explored 13 RNs’ monitoring nursing practices, found that clinical monitoring practice depends on RNs’ individual levels of professionalism, and RNs with strong professionalism were more likely to conduct, record, assess, and act in a more appropriate manner than RNs with less-developed professional awareness.

The lack of professional identity and unclear role perception among RNs, in combination with the low priority of NC, resulted in suboptimal, haphazard, unsystematic, and missing care, thus constituting a risk for patient safety and suboptimal well-being. This risk also appeared when fundamental and evidence-based NC was delegated to LPNs. Consequently, in several situations, patients received inappropriate treatment or care because of the lack of knowledge and skills in the LPNs; however, the interventions delegated to LPNs were not included in their palette of competences; rather, they were responsibilities of the RNs but were delegated due to, and legitimised by, the RNs’ experiencing a of lack of time. Lack of time was, by RNs, in general, perceived to be a major barrier for delivering NC. Nonetheless, the data indicated that time itself was not a barrier. Rather, it was the RNs’ use of time, unclearness about their role function, and low prioritisation of fundamental NC that were the barriers. In addition, they seemed unaware of the impact of NC on patient safety and recovery. Altogether, these barriers contributed to suboptimal quality and missed NC.

The lack of attention and attributed value to fundamental NC found in this study is not a local phenomenon. In recent years, increasing attention has been placed on the performance of NC due to the worldwide high prevalence of missing NC [4,17,18]. This poses a significant risk for patient safety, as systematic reviews have been reported to be associated with morbidity, LOS, increased readmission rates, and mortality [4,33].

The hospital context constituted a central barrier for RNs in our study to deliver evidence-based NC and withhold their professional identity. The daily routines were organised according to the demands of the biomedical model, and NC was invisible and delivered in the time left from the task and demands from apparently more powerful groups of HPs. According to Feo and Kitson [36], three factors contributed to the devaluation of NC, leading to unsafe clinical practices in healthcare settings: (1) RNs themselves do not acknowledge the value of NC, (2) the dominance of the biomedical model, and (3) healthcare systems do not acknowledge the value and impact of professional NC.

### 4.2. Team-Based and Organisational Barriers and Facilitators

The RNs in our study organised and structured their time and tasks to accommodate the demands of the biomedical model, turning RNs’ focus on the biomedical and treatment-related aspects. The biomedical model is characterised by attributing illnesses to a single physical cause (e.g., cellular abnormalities) independent of mental processes and the social environment [37]. Consequently, little attention has been paid to the psychological, social, cultural, spiritual, and environmental attributes of diseases. Nevertheless, alternative holistic, integrative, and biopsychological models argue that these aspects must be acknowledged and identified to achieve a successful diagnosis, high-quality treatment, patient-centred care, and therefore, optimal patient outcomes [36,37]. The biomedical model has resulted in great advances in the diagnosis and treatment of life-threatening and debilitating diseases, although the model has been criticised for contributing to the exclusion and devaluation of fundamental, patient-centred care [36]. In the present study, this organisational barrier further allowed the interdisciplinary team and nurse managers to ask RNs for biomedical information, thus turning RNs’ focus away from fundamental NC. Moreover, because of the RNs’ lack of awareness of their professional role and boundaries, their time and focus were stolen, leaving limited and random time gaps to deliver tasks related to fundamental care. This illustrated and contributed to a hierarchy where other professionals’ priorities and tasks were apparently attributed higher values than NC. The impact of the value systems in which nursing is performed was further identified in the qualitative studies of Lindhardt [38] and Kjerholdt et al. [39], who focused on the care of older medical patients. They found that the RNs were caught between their professional values and the system values embedded in the medical model and the new public management model [38,39], and this battle of values placed them in a moral dilemma, leading to powerlessness and a failure to deliver quality care.

We found that the contribution of fundamental NC to the patient’s recovery and safety was invisible and unnoticed, leading to the devaluation of NC and RNs, placing both at the bottom of the organisational hierarchy. This result is in line with many other studies [3,36,39]. The many interruptions we observed from other HPs may well be an indication of this hierarchy, implying, as it did, a lack of respect for and acknowledgment of nursing and RNs’ time and independent roles and functions. This finding is supported by previous research [33,38,39,40,41] indicating that the hierarchy has an impact on evidence-based nursing practices and that RNs experience a lack of support from physicians, as well as interdisciplinary awareness of the value and impact of evidence-based NC [41,42]. It is important to emphasise that this pattern found in our study did not appear to be based on conscious actions or foul intentions from other professionals; rather, it was a barrier embedded in the organisational culture by tradition, poor leadership, and a lack of societal and professional acknowledgment of the value and impact of nursing.

The nurse manager was identified as a key facilitator in our study, mediating a changed hierarchical culture and facilitating RNs in performing NC according to their professional values. However, it depended on the focus and competencies of the individual nurse managers, as leaders with poor leadership skills and lacking professional nursing focus contributed to the maintenance of the established hierarchy and, consequentially, to the time theft from nursing. This result agrees with previous research reporting that nurse managers’ leadership styles have a strong influence on the implementation of evidence-based practices and for providing conditions for this to happen [43], particularly when interventions targeting organisational barriers are required [44]. However, Enterkin et al. [45] evaluated the leadership skills among 36 nurse managers and found that they lacked the necessary skills to change clinical practices. A systematic review by Bianchi et al. [43] exploring the impact of leadership on evidence-based practice in healthcare settings emphasised leadership skills as crucial but not enough to develop and facilitate evidence-based nursing initiatives. Rather, to facilitate and sustain evidence-based nursing practices, nurse managers need to have a knowledge of evidence-based nursing practices and learn to address barriers to their implementation. They must understand their role in creating a supportive environment. Moreover, they must be able to create an empowering environment that includes passionate frontline managers and multifaceted effective implementation strategies at the individual, social, organisational, and leadership levels to support and facilitate RNs in performing evidence-based nursing practices, thereby changing patient care.

Hence, a new tradition must be initiated and supported by a strong nursing leadership and organisational and societal acknowledgement of the importance of nursing delivered systematically by well-trained, knowledgeable RNs with a strong professional identity. If this is not achieved, the RNs’ time and focus from NC will continue to be stolen, patients will experience missed care, and patient safety will continue to be at risk. We identified multiple complex barriers and facilitators, indicating that a multifaceted implementation strategy is needed to increase RNs’ professionalism and professional identities, thereby ensuring patients’ effective and safe treatment and care. This is in line with implementation scientists [46], who advocate tailored implementation strategies addressing local barriers and facilitators and targeting change at the individual, as well as team and organisational, level.

### 4.3. Methodological Considerations

The strength of the study was the data triangulation with focus group interviews, field observations, and individual follow-up interviews, allowing the identification of both the RNs’ own accounts of their practices and perceived barriers and facilitators for adherence to EBGs, as well as observations of their actual actions, priorities, conduct, and organisation of tasks in their everyday practice. The heterogeneity of the three involved units with different cultures, teams, and leadership styles enhanced the strength of the results. Furthermore, researcher triangulation in the data analysis enhanced the credibility of the findings. A limitation could be that the study was performed in the local context of the Danish healthcare and welfare system, which could reduce the transferability of the findings. However, we found substantial support in the international literature, indicating the general nature of our findings. Further, a few participating RNs in focus group interviews also participated in the observations and individual interviews, which could have influenced them to perform NC according to the EBGs. However, the results indicated otherwise. It must also be considered that the fact that the participants were in an unnatural situation as they were being observed may have altered their performances in a more ideal direction. However, according to the data, as well as the perception of the investigator (SE), this was seemingly not the case. On the contrary, the RNs expressed that they were comfortable with the researcher’s presence, as they knew the researcher very well and had been included in observational studies several times before. The data were collected by the first author only, and her knowledge and preunderstanding of the context may have limited her openness to new and unexpected aspects during the data collection and analyses. On the other hand, her familiarity with the context and daily routines may have released a surplus of awareness and attention to aspects of the research themes in question. This methodological concern was addressed by the researcher asking the RNs to clarify questions, presenting them with her interpretation and probing for their experiences with various situations in order to gain a deeper understanding of the observed situations.

## 5. Conclusions

Missed care and haphazard and unsystematic NC in hospital wards may be explained by the dominance of the biomedical model and a hierarchy where NC ends up at the bottom and RNs lose their professional identity. Strong, professional nurse leadership may reduce this pattern and make NC visible, thereby supporting RNs in regaining their professional identity.

## Figures and Tables

**Figure 1 healthcare-09-01524-f001:**
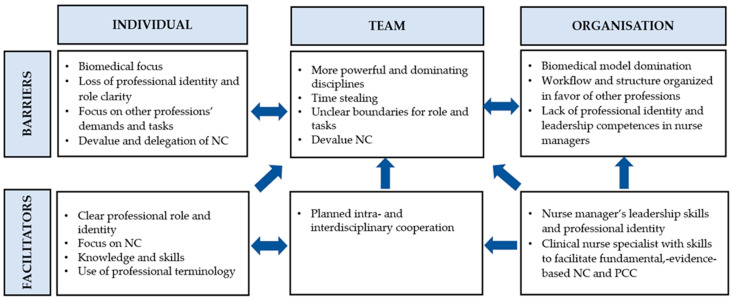
Overview of the identified barriers and facilitators at the individual, team, and organisational levels. Abbreviations: NC, nursing care and PCC, person-centred care.

**Table 1 healthcare-09-01524-t001:** Descriptions of the TDF domains.

TDF Domains	Description
Knowledge	An awareness of the existence of something
Skills	An ability or proficiency acquired through practise
Social/professional role and identity	A coherent set of behaviours and displayed personal qualities of an individual in a social or work setting
Beliefs about capabilities	Acceptance of the truth, reality or validity about an ability, talent or facility that a person can put to constructive use
Memory, attention and decision processes	The ability to retain information, focus selectively on aspects of the environment and choose between two or more alternatives
Beliefs about consequences	Acceptance of the truth, reality or validity about outcomes of a behaviour in a given situation
Environmental context and resources	Any circumstance of a person’s situation or environment that discourages or encourages the development of skills and abilities, independence, social competence, and adaptive behaviour
Social influences	Those interpersonal processes that can cause individuals to change their thoughts, feelings, or behaviours
Intentions	A conscious decision to perform a behaviour or a resolve to act in a certain way
Optimism	The confidence that things will happen for the best or that desired goals will be attained
Goals	Mental representations of outcomes or end states that an individual wants to achieve
Behavioural regulation	Anything aimed at managing or changing objectively observed or measured
Reinforcement	Increasing the probability of a response by arranging a dependent relationship, or contingency, between the response and a given stimulus

Adapted from Reference [26].

**Table 2 healthcare-09-01524-t002:** Summary table of the data.

Data Collection	Semi-Structured Focus Group Interviews	Field Observations	Individual Follow-Up Interviews
Number of data	*N* = 2	*N* = 14	*N* = 10
Number of participants	*N* = 12 RNs (*N* = 6 RNs in each group)	*N* = 14 RNs and 88 interdisciplinary HPs	*N* = 10 RNs
Duration	2 × 1.5 h	49 h	10 × 30–45 min

**Table 3 healthcare-09-01524-t003:** Overview of the main theme, themes, and subthemes.

Main Theme	‘Stolen Time’—Delivering Nursing at the Bottom of A Hierarchy		
Themes	Under the dominance of stronger paradigms	The loss of professional identity	The power of leadership
Sub-themes	Detained by the Medical Model	Filling the left-over time gaps	The mediator of culture and hierarchy
	Time-thieves	Knowledge without action	Stealing back time
		The impact of a professional terminology	

## Data Availability

The datasets generated and/or analysed during the current study are not publicly available due to their containing information that could compromise the privacy of the research participants but are available from the corresponding author upon reasonable request.

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
