# Peer review of "‘Stolen Time’—Delivering Nursing at the Bottom of a Hierarchy: An Ethnographic Study of Barriers and Facilitators for Evidence-Based Nursing for Patients with Community-Acquired Pneumonia"

_healthcare, 2021, doi:10.3390/healthcare9111524_

Round 1
Reviewer 1 Report
This is a very interesting and important area of research, especially the focus on a common but fatal diagnosis for seniors, community-acquired pneumonia. Most evidence-based guidelines (EBG) focus on acute care settings. The ethnographic approach is an excellent methodology to use for this research. I also congratulate you on looking at different systems level barriers and facilitators. I suggest inserting your over-arching research question around line 71 on page 2.
Around line 96, I think your paper will be enriched by including more description of the Theoretical Domain Framework you used and how it assisted you in your study.
Under Sample or Results, I recommend a summary table of the numbers and types of interviews and observations you did-to highlight the rigor of your work.
Under Results, your Figure 1 is excellent. Would it be possible to include more quotes to support all your sub-themes? “Stolen time” is a very innovative interpretation, since we often think solely of workload. Looking at it from your perspective is an interesting one-a real contribution to how we think about nurses’ workloads.
Author Response
Cover letter
Regarding the manuscript (Manuscript ID: healthcare-1440275), ‘Stolen Time’- Delivering Nursing at the Bottom of A Hierarchy: An Ethnographic Study of Barriers and Facilitators for Evidence-based Nursing for Patients with Community Acquired Pneumonia.
Dear Editor and Reviewer
Thank you very much for your review of the manuscript. We really appreciate the comments and have given them our full consideration. The enclosed version of the manuscript represents a revision in accordance with these comments. The changes in the manuscript are performed by using track changes. We have provided responses to your comments below.
We look forward to hearing from you.
Please see the attachment.

Reviewer 2 Report
The work contributes to the understanding of organisational, group and individual behaviours and contextual factors that influence/facilitate the adoption of evidencebased guidelines by licensed nurses in the CAP patient pathway. The research efforts made are worthwhile and the results are clear and consistent with the research aim.
Proofreading concerning typos and grammatical errors is recommended.Revision by a native speaker may improve the clarity of the document.
The theoretical framework supporting the research should be described in more detail in order to validate the instrument used in the first phase of the research (semi-structured questionnaire). The adoption of synoptic tables for both the theoretical framework and the presentation of results could significantly improve the work. The synoptic tables could also be used to emphasise the comparison of the literature with the results of the ethnographic study. Finally, it might be useful to stress the difference between endogenous and exogenous causes and to emphasise the impact of readiness factors. A clarification has to be made: were nurses involved in the validation of the implicit and explicit contents defined by the researchers? I recommend introducing the members of the research team involved in the ethnographic research during the explanation of the methodology. You should also include some consideration about the risk that nurses might behave differently during observation and the countermeasures taken to reduce this risk. Figure 1 must be correct (selected edges are visible).
I hope these suggestions will be useful.
Author Response

(The authors gave the same response as above.)
